# Identification of a Hypomorphic *FANCG* Variant in Bernese Mountain Dogs

**DOI:** 10.3390/genes13101693

**Published:** 2022-09-21

**Authors:** Katheryn Meek, Ya-Ting Yang, Marilia Takada, Maciej Parys, Marlee Richter, Alexander I. Engleberg, Tuddow Thaiwong, Rachel L. Griffin, Peter Z. Schall, Alana J. Kramer, Vilma Yuzbasiyan-Gurkan

**Affiliations:** 1Comparative Medicine and Integrative Biology Program, College of Veterinary Medicine, Michigan State University, East Lansing, MI 48824, USA; 2Department of Microbiology and Molecular Genetics, College of Veterinary Medicine, Michigan State University, East Lansing, MI 48824, USA; 3Department of Pathobiology and Diagnostic Investigation, College of Veterinary Medicine, Michigan State University, East Lansing, MI 48824, USA; 4Department of Small Animal Clinical Sciences, College of Veterinary Medicine, Michigan State University, East Lansing, MI 48824, USA; 5Veterinary Diagnostic Laboratory, College of Veterinary Medicine, Michigan State University, East Lansing, MI 48190, USA; 6College of Veterinary Medicine, Michigan State University, East Lansing, MI 48824, USA

**Keywords:** Bernese mountain dog, histiocytic sarcoma, fanconi anemia, cancer, comparative genetics

## Abstract

Bernese mountain dogs (BMDs), have an overall cancer incidence of 50%, half of which is comprised of an otherwise rare tumor, histiocytic sarcoma (HS). While recent studies have identified driver mutations in the MAPK pathway, identification of key predisposing genes has been elusive. Studies have identified several loci to be associated with predisposition to HS in BMDs, including near the *MTAP/CDKN2A* region, but no causative coding variant has been identified. Here we report the presence of a coding polymorphism in the gene encoding FANCG, near the *MTAP/CDKN2A* locus. This variant is in a conserved region of the protein and appears to be specific to BMDs. Canine fibroblasts derived from dogs homozygous for this variant are hypersensitive to cisplatin. We show this canine *FANCG* variant and a previously defined hypomorphic *FANCG* allele in humans impart similar defects in DNA repair. However, our data also indicate that this variant is neither necessary nor sufficient for the development of HS. Furthermore, BMDs homozygous for this *FANCG* allele display none of the characteristic phenotypes associated with Fanconi anemia (FA) such as anemia, short stature, infertility, or an earlier age of onset for HS. This is similar to findings in FA deficient mice, which do not develop overt FA without secondary genetic mutations that exacerbate the FA deficit. In sum, our data suggest that dogs with deficits in the FA pathway are, like mice, innately resistant to the development of FA.

## 1. Introduction

Domestic dogs have remarkably high cancer rates as compared to humans [1]. The predilection for certain dog breeds to succumb to particular cancers strongly suggests a genetic basis for breed-associated malignancies [2,3]. One of the highest breed-specific cancer associations is for histiocytic sarcoma (HS), which occurs at a rate as high as 25% in Bernese mountain dogs (BMD); this breed has an overall cancer rate of more than 50% [4]. Histiocytic sarcomas derive from the myeloid-dendritic cell lineage, and while rare in both species, have a high frequency of occurrence in certain dog breeds such as BMDs. Other breeds where HS is over-represented include the flat coated retrievers, rottweilers and golden retrievers [5,6].

Histiocytic neoplasms account for less than 1% of all cancers of the soft tissue and lymph nodes in humans [7]. In dogs and humans, they are quite heterogeneous with presentations varying from mild, localized disease to disseminated and lethal disease, involving many organ systems. While earlier studies classified the human histiocytic disorders into Langerhans cell, non-Langerhans cell and malignant histiocytosis [8], recent advances have pointed to the presence of clonal mutations of genes in the MAPK pathway in many forms of histiocytic disease [9,10,11]; these led to a revised classification of these disorders in the human in 2016 [7], and updated NCCN guidelines for classification and treatment in 2021 [12] and 2022 [13]. With the clinical use of targeted small molecule inhibitors of the key drivers, remarkable achievements in treatment with continued disease-free survival of over two years have been demonstrated [10,14,15]. First reported by our group [16,17], and followed by others [18], mutations in MAPK pathway, especially in PTPN11 and in KRAS, have also been identified as key drivers in HS in dogs, further demonstrating the significance of the dogs as excellent spontaneous translational models. We have also demonstrated the potential efficacy of inhibitors of the MAPK pathway, such as trametinib and dasatinib, in canine HS cell lines and mouse models of HS [19,20,21]. Despite these advances, key mechanistic insights into predisposition to HS have continued to elude the field.

Fanconi’s Anemia (FA) is a genetic disease in humans with a heterogeneous phenotype; the variability in phenotype is partially explained by the fact that Fanconi’s Anemia can be caused by mutations in one of at least 22 different genes [22,23,24,25,26]. The phenotypes that most commonly affect FA patients include: (1) bone marrow failure by age 40 (90%); (2) congenital birth defects (60–70%), and cancer, most commonly acute myelogenous leukemia (AML) (20%); a cancer that is derived from the monocyte/macrophage lineage. The median age of death for FA patients is 30 years. Cells from patients with FA are remarkably sensitive to DNA cross-linking agents, indicating a DNA repair function for the FA pathway. Decades of research have shown that the FA pathway functions to protect the genome during DNA replication. Briefly, if a DNA replication fork encounters a DNA cross-link, the fork is arrested. The key event of the FA pathway is dependent on an eight-protein core complex, the Fanconi anemia core (FANC) complex, which includes FANCG. This complex directs mono-ubiquitylation of the FANCD2-FANCI complex and targets the FAN1 nuclease to the DNA crosslink, “unhooking” the fork. After trans-lesion synthesis bypasses across the “unhooked” adduct, nucleotide excision repair removes it from the second strand [27,28,29]. Ubiquitylated FANCD2 promotes interaction of BRCA1 with BRCA2 (also called FANCD2) promoting the homologous recombination phase of DNA cross-link repair [30].

Recently, association of HS susceptibility to loci near the *MTAP-CDKN2A* locus on canine chromosome (CFA)11 has been shown in BMDs [31,32]. Markers at either end of this locus are in linkage disequilibrium (LD). The canine genome has much longer regions in LD (20–50 times longer than the human genome), and the genomes of certain breeds, for instance BMDs, have longer regions in LD than others [33]. Although the CDKN2 genes are well-characterized tumor suppressors, no clear impactful, coding polymorphisms have been demonstrated [31,32]. Moreover, expression of the tumor suppressor transcripts was higher in the allele associated with HS as compared to the protective allele, a counterintuitive result if differing CDKN2 expression were to explain differences in susceptibility to HS. We considered that a linked coding region polymorphism might contribute to the association of this locus with HS and examined this region for coding polymorphisms. 

We report the identification of a polymorphism in canine *FANCG* (Fanconi’s Anemia Complementation Group G, ~7 MB from the *MTAP-CDKN2A* locus, and explore its association with the development of HS or to earlier age of onset of HS, and functional studies examining the effect of this variant on DNA repair.

## 2. Materials and Methods

### 2.1. Selection of Cases

The BMD samples were part of our BMD DNA and Tissue Repository at Michigan State University collected under MSU IACUC approval (AUF #08/15-127-00). All dog owners provided written consent allowing the use of the submitted tissue and blood samples for research. All cases of HS were confirmed with histopathology or cytology and reviewed by a board-certified pathologist (TT). Germline DNA was isolated from blood from 121 dogs, ages 1–12.5, with a confirmed HS diagnosis. To explore relationship of onset of age in an unbiased manner, all dogs diagnosed with HS that were younger than 6 years of age and had blood samples in the repository (totaling 24 dogs) were included in the study. Owners of the dogs in the repository that had reached ages 9 or greater were contacted, and all were included in the genotyping in the “no cancer” group if they were confirmed to be free of cancer based on owner declaration (*n* = 66).

BMDs with Cancers other than HS: Blood was available from 21 BMDs, (ages 1–12.5 years, average age 7.9) with a diagnosis of cancer other than HS. These included 8 males and 13 females. The cancer types included: cholangiocarcinoma (*n* = 1), adenocarcinoma (*n* = 2), lymphoma (*n* = 5), carcinoma (*n* = 1), hemangiosarcoma (*n* = 2), unspecified brain tumor (*n* = 1), round cell tumor (*n* = 1), osteosarcoma (*n* = 3), sarcoma (*n* = 1), fibrosarcoma (*n* = 1), unspecified cancer (*n* = 1), and multiple cancers (*n* = 2, lymphoma and spindle cell tumor; melanoma and unspecified cancer).

Multiple breed panel: A DNA panel from various breeds of dog, established from dogs (*n* = 5 per breed) unrelated to each other for over three generations was used. The breeds represented included the greyhound, pointer, border collie, Labrador retriever, German shepherd, Scottish terrier, beagle, Doberman pinscher, Siberian husky and cocker spaniel, as well as five mixed bred dogs.

### 2.2. Isolation of DNA

DNA was extracted from blood samples banked in our laboratory as the BMD DNA and Tumor Repository. Blood samples were obtained from client-owned dogs and shipped with cold packs to MSU, where they were kept in aliquots at −80 °C. Genomic DNA was extracted from 100 µL of whole blood using the DNeasy Blood and Tissue kit (69504, Qiagen LLC., Germantown, MD, USA). For quantification, the DNA concentrations were measured by Qubit™ dsDNA HS kit (Q33230, Thermo Fisher Scientific Inc., Waltham, MA, USA) using a Qubit 2.0 fluorometer (Thermo Fisher Scientific Inc., Waltham, MA, USA).

### 2.3. Library Preparation and Whole Exome Sequencing

The SureSelect Canine Exome and SureSelect XT library preparation kits (Agilent Inc., Santa Clara, CA, USA) were used to prepare the DNA from the blood of an 8-year-old BMD affected with HS for sequencing. Paired end sequencing was carried out using Illumina HiSeq 2500 High Output Flow Cell (2 × 125 bp). Library preparation and sequencing were performed at Michigan State University Research Technology Support Facility Genomics Core. Resulting sequence data were aligned to the CanFam3.1 reference genome using Bowtie2 [34]. Variant calling was performed using Freebayes and GATK [35,36]. 

### 2.4. Genotyping Assays

DNA samples were genotyped using custom-made TaqMan SNP Genotyping Assays specific for the MTAP and FANC alleles, assay IDs were AN9HM4H, part number 4,332,077 and AHUAP1T, part number 4,332,077, respectively, in 96-well plate format. Each genotyping reaction contained 0.5 μL of 20× TaqMan Custom SNP Assay, 5 μL of TaqMan Genotyping Master Mix (4,371,355, Thermo Fisher Scientific Inc., Waltham, MA, USA) and 5 μL of genomic DNA at 2 ng/μL. For each assay, two no-template negative controls containing DNase free water, and three positive controls with known genotypes, one heterozygous, and two others (homozygous for each allele) were included. Genotyping was conducted in real-time PCR mode using a StepOnePlus or Quant3Studio PCR System (Thermo Fisher Scientific Inc., Waltham, MA, USA) following the cycling conditions instructed by the manufacturer. All assays were performed in duplicate.

### 2.5. Measurement of Height at Withers and Weight and Complete Blood Count

BMD owners who were known to be veterinarians, veterinary technicians or biomedical researchers, and whose dogs were part of the BMD DNA and Tissue Repository, were contacted for participation in this study and instructed to measure height at withers and provide weights of their dogs, and when available provide complete blood count (CBC) information from their records. In addition, Dr. Cheri Beasley and her colleagues at Cedar Creek Veterinary Clinic kindly provided weight, height at withers, and CBC information and blood samples for 15 healthy BMDs with client consent. 

### 2.6. Cell Culture, Plasmids, and Transfection

Primary fibroblasts from an HS affected BMD were derived from leftover surgical specimens from the MSU Veterinary teaching hospital. Normal fibroblasts were described previously [37]. The 293T cells were the generous gift of Dr. Kefei Yu (who obtained the cell strain from ATCC). Cells were cultured in Dulbecco’s MEM with 10% fetal calf serum, penicillin/streptomycin, and cipro (complete medium), and maintained at 37 °C with 5% CO_2_. 

### 2.7. Cas9-Mediated Gene Disruption

Cas9-targeted gene disruption was performed using methods similar to those reported by Mali et al. [38]. Briefly, oligonucleotides encoding gRNA targets spanning 20 nucleotides 5′ of the PAM site were synthesized (Integrated DNA Technologies Inc., Coralville, IA, USA), and ligated into the BSSH1 site of pCAS2A puro (Addgene, Watertown, MA, USA) that allows co-expression of a specific gRNA, Cas9, and the puromycin resistance gene. Cells were transfected with 2 µg plasmid in 200 µL Optimem (Thermo Fisher Scientific Inc., Waltham, MA, USA) and 4 µL PEI polyethylenimine (PEI, 1 ug/mL, Polysciences Inc., Warrington, PA, USA). To promote homology directed repair (to generate the two specific mutations), 120 bp oligonucleotides encoding each, as well as a silent mutation generating a novel restriction site, was co-transfected. At 48 h after transfection, cells were replated at cloning densities in media containing puromycin (1 µg/mL). Puromycin was removed after 72 h. Isolated clones were selected, and DNA isolated with DNAzol (Sigma-Aldrich, St. Louis, MO, USA) according to the manufacturer’s protocol. Clones were screened by PCR amplification using primers as listed in Appendix A, and the presence of the anticipated mutations were confirmed by TA cloning (Invitrogen, Waltham, MA, USA) and sequencing of PCR products. The 20-mers specific for each gRNA (without BSSH1 overhangs) and the sequences of primers used are detailed in Appendix A.

### 2.8. Viability Assays

Viability assays were carried out using the MTS assay (Promega Corporation, Madison, WI, USA) for 293T cells and primary fibroblast cell strains, according to the manufacturer’s directions. Five thousand cells were plated in each well of a 96-well plate, containing medium with varying concentrations of cisplatin (APP Pharmaceuticals LLC, Schaumburg, IL, USA). After 4 to 5 days of cisplatin treatment, 20 µL MTS was added; absorbance was read at 490 nm to determine relative survival, as compared to vehicle controls. 

### 2.9. Immunoblotting

The 293T cells were treated with 20 µM cisplatin, or mock treated with vehicle (0.9% NaCl) overnight. Whole cell extracts were prepared by solubilization in the following buffer: 50 mM HEPES, 150 mM NaCl, 1 mM EDTA, 5 mM MnCl2, 0.1% TritonX-100, and 300 µg/mL DNAse; after 5 min on ice, an equal volume of SDS Page buffer was added. Proteins were analyzed by electrophoresis on 6% SDS polyacrylamide gels and transferred to PVDF membranes and probed with antibody. The FANCD2 antibody was obtained from Santa Cruz (catalogue #28194). 

### 2.10. Statistical Analysis

Statistical analysis was carried out using Graph Pad Prism 9.2.0, Graph Pad Software Inc., (San Diego, CA, USA) with significance set at *p* < 0.05 for all studies. Specific tests used for each data set are indicated in the associated figure and text.

## 3. Results

### 3.1. A Coding Variant in FANCG, Unique to the Genome of BMD is Identified

Exome sequence data from an 8-year-old BMD affected with HS was examined for coding polymorphisms near MTAP/CDKN2A. A single homozygous nucleotide change located at position CFA11:51655193 in CanFam3.1, XP_038537954.1:p.(Gln649Arg), now listed as rs850607580:T > C in the European Variation Archive that would encode a Q > R substitution at position 465, NP_004620.1:p.(Gln465Arg), based on the human protein, was identified (Appendix A). The variant was found to be present in both germline and tumor DNA. As can be seen in Figure 1, this Q > R substitution is in a well-conserved residue in mammals. The alignment of the human transcript and predicted canine mRNA sequences are presented in Appendix A. The variant residue is found in position 649 and 477 of the two predicted canine transcripts. 

The structure of the canine and human *FANCG* genes is schematically depicted in Figure 2. In both species there are 14 exons, with the first exon in the dog being longer and containing two predicted start sites. The human and canine protein sequences encoded by exons 2–14 show 81% identity and 93% similarity. Appendix A further details similarities in the 5′ UTR of the human and exon 1 of the dog, indicating 64% identity at the nucleotide level even in the untranslated region of the human sequence.

### 3.2. The FANCG Variant Is Unique to BMDs

We next explored whether this allele was present in dogs from other breeds. Using a DNA panel from various breeds of dog (unrelated to each other for over three generations) available in our laboratory, and examining five dogs of each breed (55 total), we determined that all dogs in the panel were homozygous for the reference allele. The breeds represented included the greyhound, pointer, border collie, Labrador retriever, German shepherd, Scottish terrier, beagle, Doberman pinscher, Siberian husky and cocker spaniel, as well as five mixed bred dogs. Additionally, searches for variants in the European Variation Archive at the *FANCG* locus (CFA11:51652944-51659751 in CanFam3.1) yielded only a single dog (a BMD) that carried the variant allele. This particular sample was part of a whole genome sequencing project that included 590 canine samples (582 dogs of various breeds and 8 wolves, https://www.ebi.ac.uk/eva/?eva-study=PRJEB32865, accessed on 14 July 2022). We performed a further search against data from the NHGRI Dog Genome Project (genome-wide variant discovery), which contains whole genome sequencing data from 722 canids (668 domestic dogs) from a variety of breeds. From this dataset, two samples contained the *FANCG* variant of interest; when the metadata file was examined, both samples were identified to be from dogs of the Bernese Mountain Dog breed (https://www.ncbi.nlm.nih.gov/bioproject/PRJNA448733, accessed on 14 July 2022). In total, this allowed us to explore alleles in 1305 dogs, or 2610 chromosomes, with only 3 identified dogs, all BMDs, carrying the variant allele: two were heterozygous, one was homozygous for the variant allele.

### 3.3. The FANCG Variant Has a High Allele Frequency in BMDs

Using banked blood samples in our BMD DNA and tissue repository, the frequency of the variant allele was assessed in germline DNA from BMDs presenting with histiocytic sarcoma or other cancers, as well as in dogs who had reached 9 years or above and were free of cancer. The *FANCG* variant allele frequency is slightly higher in dogs with HS (41%) vs. dogs free of cancer at 9 years or above (36%), or dogs with other cancer (32%), but the differences are not statistically significant. 

We analyzed the distribution of *FANCG* genotypes across age groups in BMDs presenting with HS, to determine if allele dosage was related to early onset of disease. As can be appreciated in Figure 3, there was no preponderance of a particular genotype across different age groups or age of presentation with HS. 

We examined the age of diagnosis of HS in 121 dogs in our repository. The mean was 7.7 years with a standard deviation of 2.15 (median 7.9) at diagnosis. As 87% of the dogs with HS presented at 10 years or younger in our cohort of HS dogs, very similar to that observed in 1818 cases with a veterinarian supported diagnosis of HS in the Berner Garde database (see Appendix A), we only included dogs >10 years of age in the no cancer group to evaluate if the variant allele frequency is different in healthy dogs 10 years of age or older, as compared to affected dogs. As shown in Table 1, the variant allele frequency in 47 unaffected BMDs 10 years or older is 39%, very close to the frequency calculated from genotypes of all ages presenting with HS, 41%. There was also no statistical difference in the *FANCG* genotype distribution across the groups. 

### 3.4. FANCG and MTAP Loci Appear to Segregate Independently in BMDs

We considered that the *FANCG* variant might contribute to the association of the *MTAP* variant with HS in BMDs, which could occur if the *FANCG* locus was in linkage disequilibrium with the *MTAP* locus. We genotyped the BMDs at *MTAP* locus using the SNP at position CFA11:41185205 in CanFam3.1, which is part of the haplotype linked to HS susceptibility. The distribution of allele frequencies is presented in Table 1.

The *MTAP* SNP reference allele frequency was 53%, 78%, and 63% in the no cancer, HS, and other cancer groups, respectively. While the *MTAP* SNP allele distribution differed significantly between the HS group vs. the unaffected group (*p* < 0.0001 Fisher’s exact test) as expected with the reported linkage to HS susceptibility [31,32], there was no statistical difference in the FANCG allele distribution across the groups, and no evidence for linkage disequilibrium with *MTAP* SNP and *FANCG* variant. 

We conclude that this variant is neither absolutely required nor sufficient to promote development of HS in BMDs. However, it is possible that it may be one of the factors confounding cancer development in this breed. We explored if there were any phenotypic differences in height, weight, and hematologic profiles among dogs of different genotypes, as has been reported in Fanconi anemia patients. 

### 3.5. BMDs Homozygous for the Variant Allele Do Not Display an Overt FA Phenotype

The variant allele was initially recognized in our study of an 8-year-old, male BMD that presented with HS. Although HS induces marked effects on hematopoiesis, no manifestations of anemia were reported. Moreover, the affected individual was of expected size for a male BMD and the animal had sired numerous offspring.

We next assessed physical characteristics (height at withers and body weight) of healthy adult female BMDs greater than 2 years of age, and their *FANCG* genotype. The differences between categories were analyzed by one-way ANOVA with Dunnett’s multiple comparison test. As depicted in Figure 4, there was no difference in height or weight among dogs of different genotypes. Height, weight, and genotype information was also obtained from 13 male dogs (see Appendix A), but as there were not enough male subjects for statistical analysis for males, details are presented below only for the female BMDs. 

Complete blood counts (CBCs) were available from a subset (*n* = 15) of these unaffected dogs and included 8 heterozygous dogs, 4 dogs homozygous for the variant, and 3 dogs homozygous for the reference allele (Appendix A). There was no evidence for anemia in any of the dogs. 

### 3.6. Primary Fibroblasts from Dogs Homozygous for FANCG Variant Allele Display Increased Cisplatin Sensitivity

We next considered that the variant allele might not impact function of the FA pathway and we considered strategies to assess the functional consequence of the *FANCG* ^465^R variant on the cellular response to DNA damage. FANCG does not have an enzymatic function. Sequence analyses of the FANCG protein reveals seven tetratricopeptide repeats (TPRs) [39,40,41]. TPRs are degenerate 34 residue repeats present in tandem (3–16 repeats) that induce protein folding into right superhelical repeats. It has been hypothesized that the TPR motifs in FANCG facilitate its function as a scaffold protein in the core FANC complex that is critical to direct FANCD2 mono-ubiquitination during repair of DNA inter-strand crosslinks. Moreover, the previous mutational study [40] that defined the seven repeats in FANCG concluded that repeats 2, 5, and 7 are critical to FANCG function. TPR repeat 5 spans the human residue 465, which corresponds to the canine residues 649 or 477, depending on the transcript. The TPR repeats are highly conserved between the two species as highlighted in Appendix A. Appendix A shows the location of this residue in a three-dimensional protein model. As Fanconi anemia is a human disorder with a set of variants characterized in the literature [42], we have elected to refer to the variant in terms of the corresponding human amino acid sequence for the functional studies reported here. 

To determine whether the variant encoded by this *FANCG* allele alters the function of the FANC complex, a primary fibroblast cell strain was established from an animal homozygous for the variant allele. Cisplatin sensitivity, a surrogate cellular phenotype for defects in the FANC pathway, was assessed in these cells, as cisplatin-caused DNA lesions are repaired by the FA pathway [22,23,24] and compared to that of a fibroblast cell strain established from a normal dog (Figure 5). As can be seen, cells from the animal homozygous for the variant are substantially more cisplatin sensitive than cells from the dog with the reference *FANCG* allele.

### 3.7. The ^465^R Substitution Introduced into Human HEK293T Cells Confers Intermediate Cisplatin Sensitivity as Compared to FANCG Deficient 293T Cells

To facilitate a more complete assessment of the ^465^R variant, we utilized a CRISPR/Cas9 strategy to generate both FANCG deficient 293T cells and 293T cells harboring this specific variant. A single gRNA targeting construct was generated targeting this residue. The 293T cells were transfected with just the targeting plasmid alone, or together with a single stranded oligonucleotide to guide HDR repair that included a Q > R substitution at codon 465. Numerous clones that completely lacked a wild type PCR fragment spanning the target site were isolated from transfections including only the targeting plasmid (Figure 6A).

Introduction of an EcoNI site in the oligonucleotide directing HDR repair facilitated screening for clones including the ^465^R substitution. Clones harboring the ^456^R mutation on both alleles yield amplification products that uniformly yield the predicted restriction fragments. Two clones for each mutant were analyzed; PCR fragments were TA cloned and sequenced confirming complete loss of *FANCG* or uniform ^465^R mutation.

Cisplatin sensitivity, a surrogate cellular phenotype for defects in the FANC pathway, was assessed in these cells and compared to wild type 293T cells, as cisplatin-caused DNA lesions are repaired by the FA pathway. As can be seen (Figure 6B), the FANCG deficient and FANCG ^465^R cells are substantially more cisplatin sensitive than wild type cells. The ^465^R cells display intermediate cisplatin sensitivity. Two independent clones for each *FANCG* genotype were tested with indistinguishable results (not shown).

We conclude that the ^465^R allele represents a hypomorphic mutation. Consistent with that conclusion, mono-ubiquitylation of FANCD2 in the ^465^R cells is reduced, but not completely blocked, whereas FANCD2 mono-ubiquitylation is completely blocked in FANCG deficient cells (Figure 6C).

### 3.8. The ^465^R Variant Confers Similar Cisplatin Sensitivity as a Human FANCG Hypomorphic Variant That Confers Fanconi’s Anemia in Humans

Numerous previous studies of FA patients have ascribed hypomorphic mutations as disease associated alleles. Thus, these data present the following questions: (1) is the ^465^R allele sufficiently functional to prevent overt FA? or (2) are there intrinsic differences in humans and canines that protect canines from overt FA? The latter explanation is clearly the case in FA deficient mice, which do not develop overt FA without additional genetic changes that promote FA [22,23]. The lack of an FA phenotype in FA deficient mice has been ascribed to lifespan, and/or size differences. However, the average life span (~8 years) of BMD (relatively short because of their high cancer incidence) should be sufficient to develop symptoms of FA. Since BMDs are more similar in size to humans as compared to mice, lack of an FA phenotype cannot be ascribed to body size differences. To address this question, we studied a FANCG hypomorphic variant (L71P) that results in overt FA in humans, albeit with a less severe phenotype and later onset [43]. A similar CRISPR/Cas9 strategy was utilized as with the ^465^R mutation, except a BstEII site was introduced to provide a PCR based screen. Two clones, with the ^71^P mutation on one allele and frame shift mutations on the second allele (confirmed by sequence analysis), were tested with analogous results. As can be seen in Figure 7, the ^71^P mutation confers similar cisplatin sensitivity as the ^456^R mutation; it seems likely that both would confer FA in humans. Two independent ^71^P clones were tested with similar results (not shown), each displaying similar cisplatin sensitivity, and each just slightly more cisplatin resistant than the ^465^R mutant. A third clone with the ^71^P on one allele with a 6 bp deletion on the second allele was also tested, with similar results (not shown). These data suggest that canines, like mice, are intrinsically resistant to the development of most of the FA-associated phenotype, even in the face of defective FA function. 

## 4. Discussion

The *FANCG* variant characterized in this study is a variant unique to BMDs. The variant is in a highly conserved region. The two predicted canine transcripts indicate there may be potential differences in start codon usage in exon 1. Furthermore, the 5′ UTR of the human retains great similarity, conserving 64% of the nucleotides in exon 1 of the dog. Exons 2–14 are highly conserved, including all of the TPR regions as shown in the alignment in Appendix A, which also includes the *FANCG* variant. Initially identified in our study in a BMD with HS, the variant has not been observed in other dog breeds, including our panel of 55 dogs from various breeds, or reported in databases such as the European Variation Archive that lists variants across studies containing many different breeds, including from a study of 582 dogs [44], as well as in a whole genome study of over 700 dogs [45]. The variant does not confer susceptibility to earlier onset of cancer. Thus, this *FANCG* variant is neither sufficient nor required to confer development of HS or other tumors observed frequently in BMD. However, in functional studies, the variant confers increased cisplatin sensitivity similar to that observed with a hypomorphic L71P mutation described in a child with Fanconi anemia.

It is intriguing that Fanconi anemia patients, including those with mutations in *FANCG*, are associated with the development of myeloid leukemias [46]. Interestingly, in one these patients the driver mutation in the tumor was identified to be in *PTPN11*, in the same gene that frequently acquires somatic mutations leading to HS in BMDs [16,17]; and in two other FA patients, mutations in the tumors were in *KRAS*, also identified in HS of BMDs [17]. 

It is well established that FA-deficient mice do not succumb to the severe bone marrow failure observed in FA patients. Several explanations have been put forth (life span, body size) to explain these differences. BMDs homozygous for the “human” ^465^R variant display none of the non-cancer phenotypes associated with FA (bone marrow failure, short stature, infertility). We suggest that as with mice, dogs are intrinsically resistant to the development of FA-associated bone marrow failure. We have considered what factors are intrinsically distinct in mice and dogs compared to humans. All higher vertebrates utilize two pathways to repair DNA double strand breaks (DSBs): homologous recombination (HR) and non-homologous end joining (NHEJ). It is well established that these two pathways “compete” for repair of DSBs. As discussed above, FANCD2 facilitates the last stage of DNA cross-link repair by HR. Of note, expression of certain NHEJ factors (especially the DNA-dependent protein kinase complex, DNA-PK) is remarkably higher in primates than in all other animals [47,48]. Moreover, several studies conclude that ablation of NHEJ in FA deficient cells partially reverses the characteristic hypersensitivity to DNA cross-linking hypersensitivity associated with FA deficiency [49,50]. and progression of the FA phenotype [51]. One possibility is that higher DNA-PK levels in humans exacerbate the organismal phenotype associated with FA deficiency. We ablated DNA-PKcs in *FANCG−/−* 293T cells and observed a modest increase in cisplatin resistance consistent with these previous studies (Appendix A). However, the finding that ablation of NHEJ improves the cellular phenotype in FA deficient cells is controversial, with follow-up studies concluding that ablation of NHEJ does not impact the FA phenotype in other cellular models of FA [52,53]. 

The fact that defects in repair of DNA replication-associated DNA damage results in failed hematopoiesis is consistent with the fact that hematopoiesis requires continuous replication. Repair of replication-associated damage is highly dependent on homologous recombination, and we have shown previously that increasing levels of DNA-PK can impair HR in cell culture models of DNA repair [54]. If the high levels of DNA-PK present in humans exacerbate DNA replication-associated damage, one might expect additional differences (between humans and other animals) in the phenotypes of diseases that are caused by a failure of DNA replication during hematopoiesis. One example of phenotypic differences is the phenotype of vitamin B12 deficiency in humans vs. dogs. Vitamin B12 deficiency causes severe anemia in humans because of the depletion of nucleotide pools that results in replicative stress. Although B12 deficiency in dogs, either because of dietary deficiency or genetic defects in absorption, results in gastro-intestinal symptoms, muscle weakness, and modest effects on myelopoiesis, the severe anemia observed in humans does not develop [55,56,57]. Similarly, studies of differences in cancer susceptibility within and across species will continue to provide interesting new insight into basic biology, including successful aging. As an example, elephants, a large and long-lived species that is remarkably cancer free, have over 20 copies of the TP53 gene [58].

What is not controversial is a clear and emerging consensus that multiple genetic factors can influence the penetrance of the FA phenotype. The most compelling of these are studies from Patel and colleagues demonstrating that deficits in ethanol metabolism robustly induce a classic FA phenotype in FA deficient mice [59,60,61,62,63]. There are additional cellular changes that have the capacity to impact development of the FA phenotype, for example dysregulated TGF-β signaling [64] and MAD2 defects [65]. In sum, we conclude that the FA deficit in BMDs, as in mice, does not result in overt Fanconi anemia. 

## 5. Conclusions

Our studies have identified a unique FANCG variant in BMDs that results in a cellular phenotype of defective DNA repair, but is not sufficient or necessary for development of cancer in BMDs. Our findings point to interesting species-specific differences in the FANC core complex. Given that FA phenotypes are only apparent with multiple FA pathway mutations, or additional mutations in other pathways, further studies of other variants in the FANC genes in BMDs, and of epigenetic changes or structural and non-coding variations in regions critical to DNA repair and chromosome stability, are warranted.

## Figures and Tables

**Figure 1 genes-13-01693-f001:**
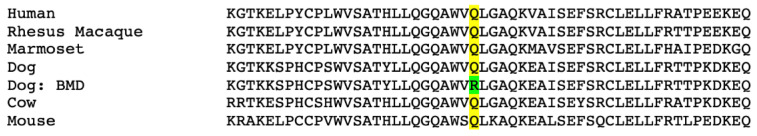
A germline-encoded FANCG polymorphism at position 465 in the human FANCG protein, identified in BMDs. In the BMD variant allele, the reference residue Q, highlighted in yellow, is substituted with R, highlighted in green. FANCG protein sequences of various mammalian species were accessed from www.ncbi.nlm.nih.gov, 4 April 2022. FANCG proteins from multiple mammalian species were aligned for comparison using Clustal Omega software (http://www.clustal.org/omega, accessed on (4 April 2022)).

**Figure 2 genes-13-01693-f002:**
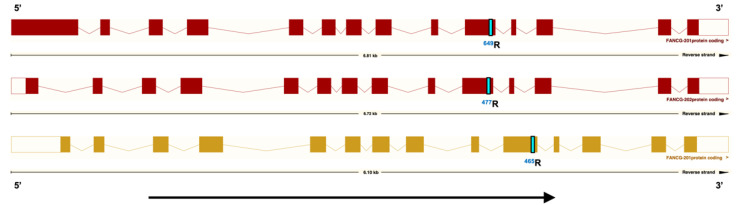
Comparison of *FANCG* gene structure between dog and human transcripts taken from www.ensembl.org, accessed on 4 April 2022, and modified as detailed here. The transcripts colored in red (■) are from the dog and were assembled using the CanFam3.1 reference genome (*FANCG* transcripts 201 and 202, respectively). The transcript colored in gold (■) is from the human and was assembled using the GRCh38 reference genome (*FANCG* transcript 201). In each transcript, the variant which is the focus of this study is highlighted in blue at its respective codon on exon 10 of the *FANCG* gene.

**Figure 3 genes-13-01693-f003:**
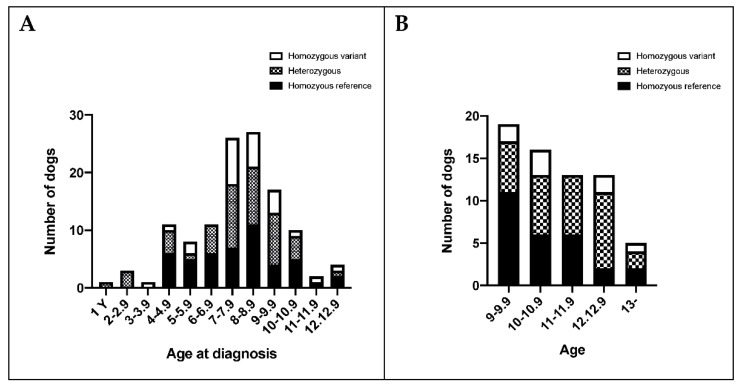
Distribution of *FANCG* genotypes across age groups in BMDs: (**A**) Dogs affected with HS across all ages, (**B**) Dogs 9 years or older and not affected with cancer.

**Figure 4 genes-13-01693-f004:**
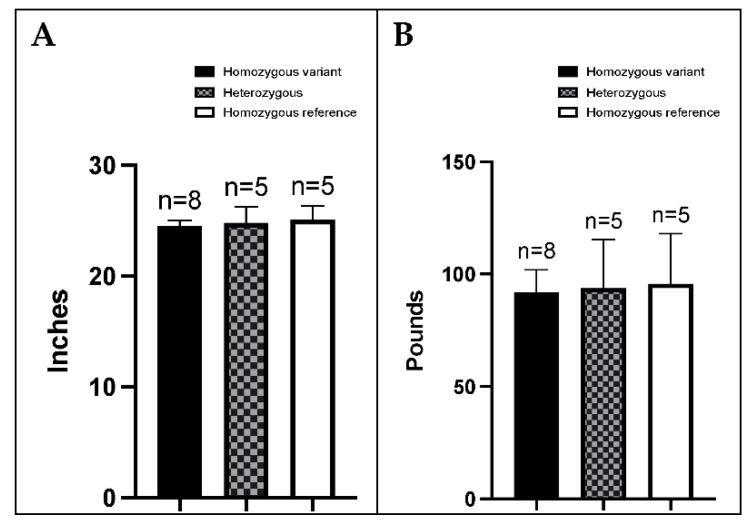
Distribution of genotype and (**A**) height at withers and (**B**) weight in healthy female adult BMDs, over 1 year of age. Dogs that are homozygous for the variant (*n* = 8, ages 1.1–10.3 years), heterozygous, (*n* = 5, ages 4.8–10.8 years), and homozygous for the reference (*n* = 5, ages 3.8–9.6 years) *FANCG* alleles are indicated by the solid black bars, the checkered bars and the solid white bars, respectively.

**Figure 5 genes-13-01693-f005:**
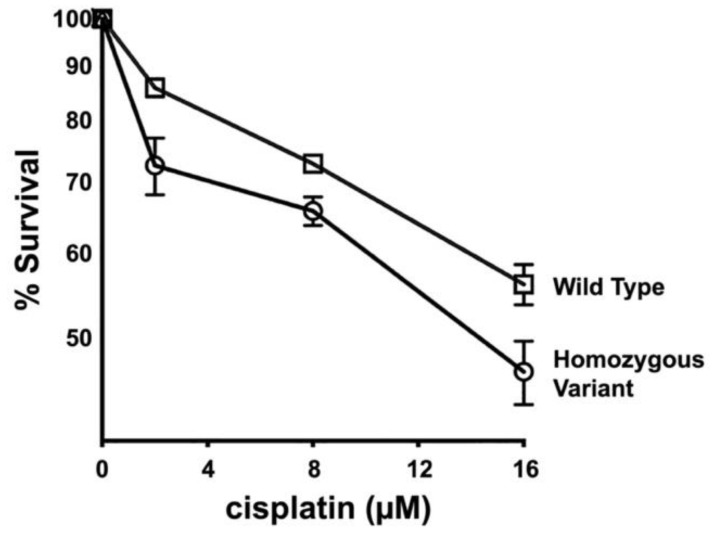
Fibroblasts derived from a dog homozygous for the variant are hyper-sensitive to cisplatin. Cultured fibroblasts from a normal dog and a dog homozygous for the variant were tested for cisplatin sensitivity. Cell viability was assessed by MTS staining and is expressed as percent survival of untreated controls. Assay is representative of five independent experiments, each performed in triplicate. Error bars represent SEM.

**Figure 6 genes-13-01693-f006:**
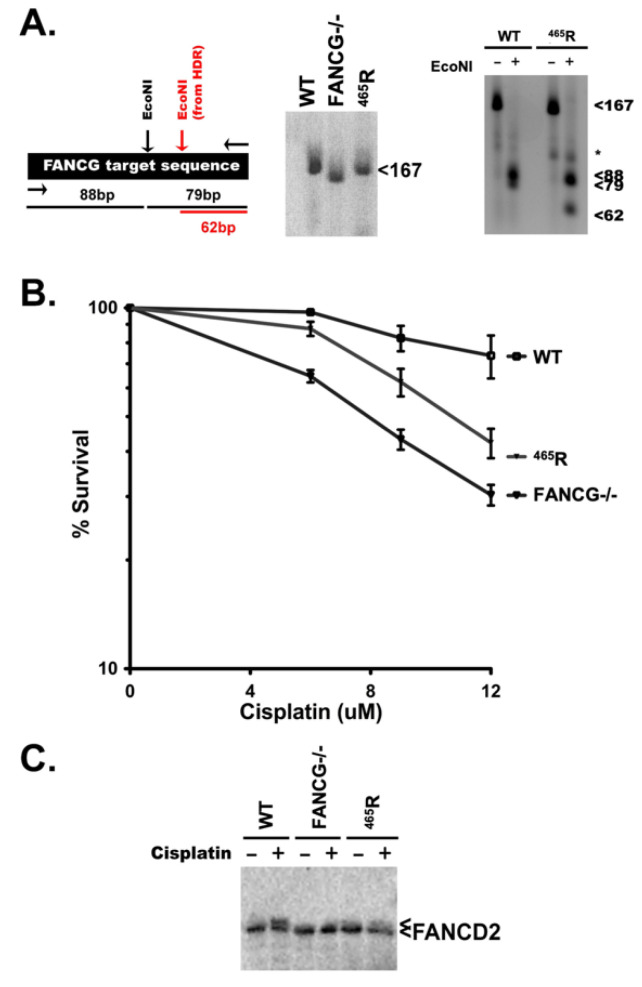
The 293T cells expressing ^465^R have intermediate cisplatin sensitivity and have diminished capacity to mono-ubiquitinate FANCD2. (**A**) Left, CRISPR/Cas9 strategy to introduce a ^465^R substitution into the human *FANCG* gene in 293T cells. Middle, PCR amplification of DNA from wild type cells, *FANCG* deficient cells, or cells with a homozygous knock-in of the ^465^R substitution. Right, EcoNI digest of PCR products. (**B**) Wild type 293T cells, FANCG−/−, and homozygous ^465^R cells were tested for cisplatin sensitivity. Cell viability was assessed by MTS staining and is expressed as % survival of untreated controls. Assay is representative of eight independent experiments, each performed in triplicate. Error bars represent SEM. (**C**) 293T cells were treated (or not) overnight with 20 μM cisplatin. Whole cell extracts were analyzed by immunoblotting for FANCD2.

**Figure 7 genes-13-01693-f007:**
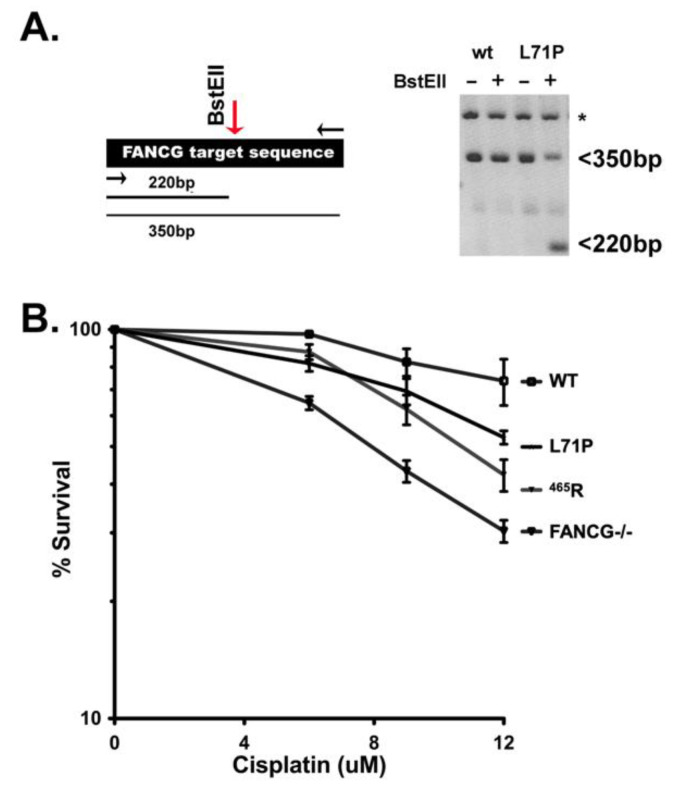
The 293T cells expressing ^71^P have intermediate cisplatin sensitivity and are slightly more resistant than cells expressing ^465^R. (**A**) Left, CRISPR/Cas9 strategy to introduce a L71P substitution into the human *FANCG* gene in 293T cells. Right, PCR amplification of DNA from wild type cells, or cells with a heterozygous L71P knock-in either undigested or digested with BstEII; sequence analyses of BstEII resistant allele of the clone analyzed revealed only out of frame insertion mutants. Similarly, one other heterozygous clone studied contained a frame shift deletion mutant. A third clone heterozygous L71P clone analyzed had a 6bp frame on the second allele; cisplatin resistance of this clone was indistinguishable from the other two heterozygous knock-ins. (**B**) Wild type 293T cells, *FANCG−/−,* homozygous ^465^R cells, and heterozygous L71P cells were tested for cisplatin sensitivity. Cell viability was assessed by MTS assay and is expressed as % survival of untreated controls. Assay is representative of eight independent experiments, each performed in triplicate. Error bars represent SEM.

**Table 1 genes-13-01693-t001:** *FANCG* allele and genotype distribution in BMDs.

Groups	Age Range (Years)	*FANCG* Variant Allele Frequency (%)	*FANCG* Genotype		
			Ref/Ref *n* (%)	Ref/Var *n* (%)	Var/Var *n* (%)
No Cancer (*n* = 47)	>10	39	16 (34)	25 (53)	6 (13)
HS (*n* = 121)	1–12.5	41	47 (39)	49 (40)	25 (21)
Other Cancers (*n* = 21)	3–12	33	10 (48)	8 (38)	3 (14)
All Cancers (*n* = 142)	1–12.5	40	57 (40)	57 (40)	28 (20)

## Data Availability

All data reported are available in the body and Appendix A of this manuscript.

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
