# Peer review of "Identification of a Hypomorphic FANCG Variant in Bernese Mountain Dogs"

_genes, 2022, doi:10.3390/genes13101693_

Round 1

Reviewer 1 Report

The paper is overall well written and tackles a fascinating and intriguing topic with a challenging complexity very different from a simple inheritance.
I think the conclusion drawn are supported by the data.
Nonetheless, I have concerns regarding the lack of details about the healthy dogs. In exploring the segregation of such a variant, we cannot just be content with it's distribution on a case/control basis. The number of the healthy dog could be increased and more importantly, their relatedness with the cases clarified, especially in the methods.

Specific comments

The authors could consider trimming the introduction, and merge few sentences. As an example, the information in 48-50 is also partially repeated later in the same introduction, where it should be probably moved.

On a similar note, the (important) 140-146 passage is awkwardly taped on – it need to be better integrated with the text. Is the discussion a better place?

202: “PCR strategies” sounds, in my opinion, too vague.

214: “treated (or not)”  is inelegant and glosses over what counts as “untreated” for the untreated cells. 0 uM solution? Nothing?

222: the authors brought on the spotlight a very interesting variant, but could they elaborate on why they decided to just explore coding regions, especially in light of the type of variants they were looking for? Are regulatory non-coding variants out of question? A posteriori, this was fruitful, but what about the experimental design? Is there any further analysis palnned for the future?

227: The author should consider to elaborate more on the impact beyond the AA level of conservation in mammals.

255-262: it’s unclear to me how the database of BMD was formed. Are the healthy dogs randomly selected? Or they are relatives of the affected ones? Shouldn’t the author search for the variant in a larger healthy BMD population? If this is the case, please specify.

378-385: the tone here is more pertinent to a Discussion – if possible, adjust this passages.

Author Response

Dear Reviewer:

We thank you for your thoughtful comments and useful feedback. We have revised the manuscript and think we have answered all of your concerns.

Please note that Supplementary Material was included at the time of the initial submission, and roles of authors were also delineated but, these apparently were not provided to you in the original submission. We have communicated this to the editors, and so Supplemental Material and authors contributions will be available in the revised version. The current submission has a revised abstract, a shorter introduction, discussion, and conclusions sections, and updated supplementary materials.  We have also included additional sections to the manuscript, on the exome sequencing and on statistical analysis.

Comment 1: The paper is overall well written and tackles a fascinating and intriguing topic with a challenging complexity very different from a simple inheritance.
I think the conclusion drawn are supported by the data.
Nonetheless, I have concerns regarding the lack of details about the healthy dogs. In exploring the segregation of such a variant, we cannot just be content with its distribution on a case/control basis. The number of the healthy dog could be increased and more importantly, their relatedness with the cases clarified, especially in the methods.

Response 2: Thank you for the thoughtful comments. We have revised the manuscript to better describe the subjects studied. As you will note, given that the predisposition to cancer is widespread in the breed, we chose to study the effect of the FANCG genotype across age groups, in the HS affected population, and in controls that were reported to be free of cancer at a relatively advanced age, and studied all dogs >9 yrs of age from which blood was available for genotyping. But, as you will see, we used an even more stringent cut off and considered dogs >10yrs of age and free of cancer as the no-cancer controls for the statistical analysis in Table 1. Supplementary Figure 1A and 1B displays the age of diagnosis in the study population of 121 dogs and that in all cases (n=1818) reported with a confirmed diagnosis in the Berner Garde Database, respectively, showing that our study population reflects that of the BMDs with HS at large.

Specific comments

Comment 2: The authors could consider trimming the introduction, and merge few sentences. As an example, the information in 48-50 is also partially repeated later in the same introduction, where it should be probably moved.

Response 2: Thank you for pointing this out. The redundancy has been eliminated and the introduction considerably trimmed.

Comment 3: On a similar note, the (important) 140-146 passage is awkwardly taped on – it need to be better integrated with the text. Is the discussion a better place?

Response 3: This section has been edited and the content integrated into the discussion.

Comment 4: 202: “PCR strategies” sounds, in my opinion, too vague.

Response 4: This has been revised to be more specific.  

Comment 5: 214: “treated (or not)” is inelegant and glosses over what counts as “untreated” for the untreated cells. 0 uM solution? Nothing?

Response 5: This has been revised to indicate that the mock treatment was 0.9% NaCl.

Comment 6: 222: the authors brought on the spotlight a very interesting variant, but could they elaborate on why they decided to just explore coding regions, especially in light of the type of variants they were looking for? Are regulatory non-coding variants out of question? A posteriori, this was fruitful, but what about the experimental design? Is there any further analysis planned for the future?

Response 6: Thank you for this. The exome sequencing that revealed this variant was one of our earlier attempts to gain more insight into genomic alterations in HS. You are certainly correct, that non-coding variants should also be explored, and indeed we are carrying out more genome wide studies in normal and tumor samples and hope to report on those as soon as we can. We include such studies as future goals in the conclusion of the revised manuscript.

Comment 7: 227: The author should consider to elaborate more on the impact beyond the AA level of conservation in mammals.

Response 7: We are not sure what the reviewer had in mind- but the study is focused on understanding the impact of this amino acid substitution.

Comment 8: 255-262: it’s unclear to me how the database of BMD was formed. Are the healthy dogs randomly selected? Or they are relatives of the affected ones? Shouldn’t the author search for the variant in a larger healthy BMD population? If this is the case, please specify.

Response 8: We have been collaborating with the Bernese Mountain Dog fancy to establish a DNA and Tumor Repository for over 10 years. As a “healthy” population does not infer a non-susceptible population when cancer frequency is so high, we sought to delineate the role of the variant by studying dogs with diagnosis that we could verify, for which we had samples. We have revised the description of the samples to make this clear.

Comment 9: 378-385: the tone here is more pertinent to a Discussion – if possible, adjust this passages.
Response 9: We have made this adjustment.

Reviewer 2 Report

The authors identified a missense variant in the FANCG gene in Bernese Mountain Dogs ("Q465R"). The mutant allele segregates in Bernese Mountain Dogs, but appears to be absent from other dog breeds. The authors experimentally demonstrated that the mutant FANCG allele leads to a higher sensitivity vs cis-platin in cell culture and appropriately claim the discovery of a hypomorphic FANCG allele. This is an interesting result. However, in its current state, the manuscript is overly long. The authors did not find any evidence that this variant is functionally related to histiocytic sarcoma or cancer in general. The authors actually confirmed that geneotypes at this variant are not associated with HS, in contrast to markers at the MTAP/CDKN2A locus that are on the same chromosome. I therefore propose that the manuscript should be shortened and more focused on the actual findings before publication.

Specific comments:

(1) The text of the manuscript should be decisively shortened. The manuscript conatins an overly long introduction about histiocytic sarcoma in Bernese Mountain Dogs, which is largely irrelevant to the study. The manuscript also contains several redundancies that should be revised (e.g. lines 50-54 are redundant with lines 76-79 to give just one example).

(2) Abstract: The abstract is misleading and must be revised. The mentioning of the MTAP/CDKN2A association with HS and subsequent discovery of the linked FANCG missense variant implies that the authors propose the FANCG variant as a candidate genetic risk factor for HS. However, the targeted association study clearly refuted this hypothesis. Therefore, the missing association of the FANCG variant to HS must be explicitly stated in the abstract! The abstract should focus much more on the actual results of the study. The "introduction to the abstract" (lines 26-31) raises the expectation that this study will reveal a new "key predisposing gene" for HS. This is clearly not the case. The results of this study are quite interesting, but the authors massively oversell them, which left me somewhat disppointed after reading the entire manuscript.

(3) Lines 113-117. The explanation of LD needs revision. LD is not a disease that "affects" the canine genome. LD within dog breeds is longer than in humans.

(4) Lines 123-146: This long section of the introduction contains a short version of results and discussion. Consider to replace this section by stating the aims of the study in one sentence and save results and discussion for the appropriate chapters.

(5) Lines 149-156: Explicitly state how many HS cases and how many HS controls were used for the study.

(6) Lines 175-182: On how many dogs were these phenotypes measured?

(7) Lines 184: Age and sex of the HS affected BMD at surgery? Age, sex and breed of the normal control dog used for figure 4?

(8) Lines 221-227: Correct variant designation according to HGVS nomenclature rules must be given for the variant (see http://varnomen.hgvs.org/).

(9) Lines 255-290. The methodology (statistics) for the association analyses is missing. This should be added to the mehtods chapter. In the results, p-values should be given. I suggest to call the two different alleles Ref and Alt (instead of Ref and Var). Table 1 contains samples from BMDs with "other cancers". These must be defined in the methods section (chapter 2.1).

(10) The methods specified that HS cases were younger than 6 years, table 1 states that HS cases ranged from 1-12.5 years, the results state that mean age at diagnosis was 7.7 years. This is clearly conflicting and needs to be thoroughly revised. How reliable is a study that uses 9 years as age-cutoff for the controls, when HS can develop in dogs as old as 12.5 years???

(11) Figure 3: Can you please display the n for each column within the figure (in addition to the information in the legend)? The genotypes could easily be indicated on the X-axis instead of using an extra inset within the figure.

(12) Discussion: The authors should briefly discuss the limitations of using cis-platin sensitivity of cultured fibroblasts or 293 cells as a proxy for the FA phenotype in vivo.

(13) Line 446: Animals or mammals?

(14) Line 452: I find it very disappointing, if experimental data are not shown. This is not in line with Gene's policies to make underlying raw data available. The data must be shown either in the main manuscript or as a supplementary figure.

(15) Line 462: Humans are an animal species. Please replace the "animal" with another term.

(16) Conclusion: Please restrict the conclusion to a very brief summary of the findings of your study. The first sentence of the conclusion reads very well and may be complemented with one more sentence detailing more results of the study. It is inappropriate for the conclusion to cite/discuss additional literature or to make vague statements about future research that has not taken place. Please delete lines 483-493. Please do not oversell your results. I really don't see any evidence whatsoever that the detected variant has any functional relevance for cancer.

(17) Are there any supplementary materials? This section contains only the default exampes from the manuscript template.

(18) The author contributions have not been specified.

(19) The data availability statement must be revised to be consistent with the supplemenatry material declaration. An accession number for the exome sequence data of the BMD dog that was used to identify the variant must be given and these data must be made publicly available.

(20) Have these exome sequence data been reported in another study? If so, this should be briefly mentioned in the methods and the apporpriate reference should be given. If the sequence data have not been used in other studies, the methodology of the exome sequencing must be described in the methods.

Author Response

Dear Reviewer:

We thank you for your thoughtful comments and useful feedback. We have revised the manuscript and think we have answered all of your concerns.

Please note that Supplementary Material was included at the time of the initial submission, and roles of authors were also delineated but, these apparently were not provided to you in the original submission. We have communicated this to the editors, and so Supplemental Material and authors contributions will be available in the revised version. The current submission has a revised abstract, a shorter introduction, discussion, and conclusions sections, and updated supplementary materials.  We have also included additional sections to the manuscript, on the exome sequencing and on statistical analysis.

Overview:

The authors identified a missense variant in the FANCG gene in Bernese Mountain Dogs ("Q465R"). The mutant allele segregates in Bernese Mountain Dogs, but appears to be absent from other dog breeds. The authors experimentally demonstrated that the mutant FANCG allele leads to a higher sensitivity vs cis-platin in cell culture and appropriately claim the discovery of a hypomorphic FANCG allele. This is an interesting result. However, in its current state, the manuscript is overly long. The authors did not find any evidence that this variant is functionally related to histiocytic sarcoma or cancer in general. The authors actually confirmed that genotypes at this variant are not associated with HS, in contrast to markers at the MTAP/CDKN2A locus that are on the same chromosome. I therefore propose that the manuscript should be shortened and more focused on the actual findings before publication.

Response:  Thank you for these comments. We have extensively edited the current submission which is much shorter and hopefully more concise.

Specific comments:

Comment (1) The text of the manuscript should be decisively shortened. The manuscript contains an overly long introduction about histiocytic sarcoma in Bernese Mountain Dogs, which is largely irrelevant to the study. The manuscript also contains several redundancies that should be revised (e.g. lines 50-54 are redundant with lines 76-79 to give just one example).

Response (1): The text of the manuscript has been shortened, especially in the introduction and the discussion, and the redundancies removed.

Comment (2) Abstract: The abstract is misleading and must be revised. The mentioning of the MTAP/CDKN2A association with HS and subsequent discovery of the linked FANCG missense variant implies that the authors propose the FANCG variant as a candidate genetic risk factor for HS. However, the targeted association study clearly refuted this hypothesis. Therefore, the missing association of the FANCG variant to HS must be explicitly stated in the abstract! The abstract should focus much more on the actual results of the study. The "introduction to the abstract" (lines 26-31) raises the expectation that this study will reveal a new "key predisposing gene" for HS. This is clearly not the case. The results of this study are quite interesting, but the authors massively oversell them, which left me somewhat disappointed after reading the entire manuscript.

Response (2) The abstract has been edited to clearly state that this variant is not sufficient nor necessary for the development of HS. (We also share your disappointment, but, that’s what the data show). 

Comment (3) Lines 113-117. The explanation of LD needs revision. LD is not a disease that "affects" the canine genome. LD within dog breeds is longer than in humans.

Response (3) This has been appropriately revised.

Comment (4) Lines 123-146: This long section of the introduction contains a short version of results and discussion. Consider to replace this section by stating the aims of the study in one sentence and save results and discussion for the appropriate chapters.

Response (4) Done. Thank you.

Comment (5) Lines 149-156: Explicitly state how many HS cases and how many HS controls were used for the study.

Response (5) This has been clarified

Comment (6) Lines 175-182: On how many dogs were these phenotypes measured?

Response (6): This has been clarified and additional data on 13 male dogs included in Supplemental Figure S3.

Comment (7) Lines 184: Age and sex of the HS affected BMD at surgery? Age, sex and breed of the normal control dog used for figure 4?

Response (7) These are now included.

Comment (8) Lines 221-227: Correct variant designation according to HGVS nomenclature rules must be given for the variant (see http://varnomen.hgvs.org/).

Response (8) We revised this section accordingly.

Comment (9) Lines 255-290. The methodology (statistics) for the association analyses is missing. This should be added to the methods chapter. In the results, p-values should be given. I suggest to call the two different alleles Ref and Alt (instead of Ref and Var). Table 1 contains samples from BMDs with "other cancers". These must be defined in the methods section (chapter 2.1).

Response (9) A section on Statistics has been added. Further information is also included on the BMDs with other cancers as well as BMDs with HS. We appreciate the suggestion to call the variant allele, alternate allele, but, this allele to us appears as a variant allele, and variant is a term universally used, as in the European Variation Archive (https://www.ebi.ac.uk/eva/)  database for variants in populations.

Comment (10) The methods specified that HS cases were younger than 6 years, table 1 states that HS cases ranged from 1-12.5 years, the results state that mean age at diagnosis was 7.7 years. This is clearly conflicting and needs to be thoroughly revised. How reliable is a study that uses 9 years as age-cutoff for the controls, when HS can develop in dogs as old as 12.5 years???

Response (10): We regret that the wording was confusing, and also that our Supplementary material was not included with the version provided to the reviewers. We made a specific point of including “all” dogs with HS that were available in our repository so as to prevent a selection bias on the younger dogs. In a parallel manner, and recognizing as the reviewer did, the difficulty of choosing dogs as “normal”, we included all dogs >9yrs of age to still be heathy in our analysis, but, only used dogs >10 yrs of age in the statistical analysis in Table 1, with the proviso that (as would have been seen in our supplementary Figure 1, and now with supplementary Figure 1a and 1b, nearly 90% of the BMDs with HS present by 10 years of age.

Comment (11) Figure 3: Can you please display the n for each column within the figure (in addition to the information in the legend)? The genotypes could easily be indicated on the X-axis instead of using an extra inset within the figure.

Response (11) We have added the n within the figure and left the inset as we think it is easier for the reader.

Comment (12) Discussion: The authors should briefly discuss the limitations of using cis-platin sensitivity of cultured fibroblasts or 293 cells as a proxy for the FA phenotype in vivo.

Response (12) Cellular sensitivity is a well-accepted proxy for the cellular phenotype for FA deficiency.

Comment (13) Line 446: Animals or mammals?

Response (13) Animals is the correct term here.

Comment (14) Line 452: I find it very disappointing, if experimental data are not shown. This is not in line with Gene's policies to make underlying raw data available. The data must be shown either in the main manuscript or as a supplementary figure.

Response (14) The data now included in Supplementary Figure S4.

Comment (15) Line 462: Humans are an animal species. Please replace the "animal" with another term.

Response (15) This has been revised to indicate “other animals”.

Comment (16) Conclusion: Please restrict the conclusion to a very brief summary of the findings of your study. The first sentence of the conclusion reads very well and may be complemented with one more sentence detailing more results of the study. It is inappropriate for the conclusion to cite/discuss additional literature or to make vague statements about future research that has not taken place. Please delete lines 483-493. Please do not oversell your results. I really don't see any evidence whatsoever that the detected variant has any functional relevance for cancer.

Response (16) The Conclusions section has been revised. While we do not see any association of this allele with cancer in BMDs, we do see functional consequences of this allele at the cellular level.

Comment (17) Are there any supplementary materials? This section contains only the default examples from the manuscript template.

Response (17) As noted above, supplementary materials were submitted by us at the initial submission, but they were not provided to the reviewers. We have contacted the editors about this, and they have confirmed that supplementary material was uploaded but not provided to the reviewers and that they will make sure to provide it in the revised version.

Comment (18) The author contributions have not been specified.

Response (18) The author contributions were specified in the original submission, but, were not appropriately reflected. The editors were alerted to this.

Comment (19) The data availability statement must be revised to be consistent with the supplementary material declaration. An accession number for the exome sequence data of the BMD dog that was used to identify the variant must be given and these data must be made publicly available.

Response (19) Since this study was limited to exploring only one variant in an interesting genomic region, we did not originally think it appropriate to include the whole exome data. However, we will upload the exome data to the Sequence Read Archive upon acceptance of the manuscript and make it available.

Comment (20) Have these exome sequence data been reported in another study? If so, this should be briefly mentioned in the methods and the appropriate reference should be given. If the sequence data have not been used in other studies, the methodology of the exome sequencing must be described in the methods.

Response (20) The exome data have not been reported in another study. A section on exome sequencing has now been added to the methodology.

Round 2

Reviewer 1 Report

I think the authors addressed most of my concerns, but I have two additional comments.

An answer to "
Response 7: We are not sure what the reviewer had in mind- but the study is focused on understanding the impact of this amino acid substitution."

When any significant, sequence-changing AA variant is detected, often it is analyzed in light of its position in the context of the protein sequence. Proteins have functional domains and other relevant sequences/sites, and the authors glossed over this once the information that the AA is conserved was shown. Where is the variant positioned in light of the known domains/loci in the protein? I'd also argue such an assessment could be helpful in the implementation of specific functional assays.

Secondly, I think the authors did not completely address the question concerning the relatedness. Comment 8. 

Author Response

Thank you for your thoughtful review.

Please find attached our point by point response and the revised manuscript. 

Reviewer 2 Report

The revised manuscript (first revision) has greatly improved with respect to the original version. However, there are still some issues that need further revisions. I apologize for bringing up some additional points that I have not seen during the first round of reviews.

(1)

The authors use entire sentences containing explicit claims as their subheadings for the results. Phrases such as "3.4. FANCG and MTAP loci appear to segregate independently in BMDs" are inappropriate as they contain an interpretation of the data. I suggest to revise all subheadings. My personal preference would be to use subheadings that describe what has been done (e.g. variant identification, association analysis, etc.) rather than specific claims. Claims should be stated in the discussion.

(2)

Chapter 3.1.: I assume that the FANCG variant was not the only coding variant "near MTAP/CDKN2A" in the studied dog. The authors should state the total number of detected coding variants and give a supplementary table listing all detected coding variants.

(3)

Line 270: It is inappropriate to give a human mRNA reference sequence for a canine protein variant. The correct designation for the human variant would be NP_004620.1:p.(Gln465Arg). I think that the homologous canine variant might be designated as e.g. XP_038527954.1:p.(Gln649Arg). This then immediately raises the questions why the canine protein has a different and much longer N-terminus than the human protein. Are there different isoforms? Are the differences possibly due to incorrect annotation? This should be briefly explained in this chapter (or in a longer section in the discussion). Please familiarize yourself with current variant nomenclature rules (Den Dunnen et al. (2016) HGVS recommendations for the description of sequence variants: 2016 update. Hum.Mutat. 25: 37: 564-569).

(4)

Table 1: The authors have not understood my comment with respect to the terms variant and allele (original comment no. 9). A variant has more than one allele! The alleles at a biallelic variant are normally termed "reference" and "alternate". Please either use "ref" and "alt" or "wt" and "mut" as abbreviations for the two alleles at a biallelic variant. The terms variant and allele are not synonymous or interchangeable!

(5)

Lines 388, 392, 395, 431, 439, 494: It is incorrect to use an (abbreviated) variant designation to specify an allele. I suggest to use  "465R allele" or 465R allele. (The confusing designations of variants and alleles should be consistently revised throughout the manuscript. This also concerns the L71P variant in chapter 3.8 and some text in the supplementary files)

(6)

Line 482: Delete "absolutely" !!!

(7)

I consider the data availability statement inacceptable. Raw data must be made publicly available prior to publication and the accession number(s) must be stated in the manuscript. I leave it up to editorial discretion whether this pont is enforced. However, the current statement does not conform to an open science policy.

(8)

Figure S3: "(B) weight in healthy female male BMDs ..." ??? "Homozygous variant" should be replaced by homozygous alternate or something similar.

Author Response

Thank you for your thoughtful comments. 

Please find attached our point-by-point response to your comments. 
